# Two-Phase Evaluation of a Community-Based Lifestyle Intervention for Palestinian Women in East Jerusalem: A Quasi-Experimental Study Followed by Dissemination

**DOI:** 10.3390/ijerph17249184

**Published:** 2020-12-09

**Authors:** Nisreen Agbaria, Maha Nubani-Husseini, Raghda A. Barakat, Elisheva Leiter, Keren L. Greenberg, Mayada Karjawally, Osnat Keidar, Milka Donchin, Donna R. Zwas

**Affiliations:** 1Linda Joy Pollin Cardiovascular Wellness Center for Women, Division of Cardiology, Hadassah University Medical Center, Jerusalem 91120, Israel; m_nubani@hotmail.com (M.N.-H.); ElishevaL@hadassah.org.il (E.L.); kereng@hadassah.org.il (K.L.G.); mkp2812@gmail.com (M.K.); donnaz1818@gmail.com (D.R.Z.); 2The Nutrition Division, The Israeli Ministry of Health, Jerusalem 91011002, Israel; rado_b82@hotmail.com; 3Braun School of Public Health and Community Medicine, The Hebrew University of Jerusalem-Hadassah, Jerusalem 91120, Israel; osnat.keidar@insel.ch (O.K.); milka@hadassah.org.il (M.D.)

**Keywords:** diabetes prevention program (DPP), healthy lifestyle, community-based intervention, Arab women, East Jerusalem

## Abstract

Palestinian-Arab women are at increased risk of cardiovascular disease due to high prevalence of diabetes and other risk factors. The current study investigates the effectiveness of an intensive group-based intervention on lifestyle habits that can prevent diabetes and cardiovascular disease. To that end, we conducted a group-based intervention based on the diabetes prevention program in two consecutive phases. The first phase consisted of a quasi-experimental study and the second phase included community-wide dissemination, with a 6-month follow-up. Findings from the quasi-experiment indicate increased consumption of fruit, vegetables and whole grains, weight reduction (−2.21 kg, *p* < 0.01), and a significant increase in the average daily steps in the intervention group (from 4456 to 6404). Findings from the dissemination indicate that average daily vegetables consumption increased from 1.76 to 2.32/day as did physical activity and average daily steps (from 4804 to 5827). There was a significant reduction in blood pressure, total cholesterol and LDL. These gains were sustained over 6 months following the intervention. This community-based, culturally adapted, health-promotion intervention led to improved nutrition and physical activity which were maintained after 6 months. Collaboration with community centers and local community partners created an effective channel for dissemination of the program to pre-clinical individuals.

## 1. Introduction

There has been a marked increase in the burden of non-communicable diseases (NCDs) in the Arab world, including diabetes and cardiovascular diseases (CVD) [1]. With rapid urbanization, increased sedentary lifestyle and obesity contribute to these trends [2]. Palestinian-Arab women living in East Jerusalem (EJ) comprise a particularly high-risk population. In 2006, the prevalence of diabetes in EJ was 41% among men and 64% among women, compared to 29% among Jewish men and 42% among Jewish women in Israel [3]. Some of this discrepancy is due to reduced health care utilization [4], dietary differences, and a high prevalence of obesity among Palestinian-Arab women [5]. Arab women in EJ are at higher risk for physical inactivity compared to both their Jewish counterparts and Arab men [6]. Additionally, diabetic patients living in EJ report receiving less counselling regarding nutrition, physical activity (PA) and foot care than their Jewish counterparts [7].

Reduction in behavioral risk factors such as smoking, physical inactivity, and low fruit and vegetable consumption has been associated with decreased risk of death from NCDs [8]. The World Health Organization (WHO) has recognized the need for effective evidence-based interventions to promote healthy nutrition and PA to reduce the burden of NCDs [9]. Interventions aimed at reducing these risk factors at the community level may be the most cost-effective methods for prevention of major NCDs [10]. Community-based programs are found to improve nutritional behaviors, weight loss and prevent diabetes [11]. A systematic review of community-based interventions targeting secondary prevention of CVD found that these interventions are effective in promoting healthy behaviors including PA [12]. Similarly, a meta-analysis of 36 studies found that multifactorial lifestyle-interventions can drastically reduce multiple risk indicators such as blood pressure, cholesterol and waist circumference in 6 months, and can increase PA at 12-month follow-ups [13]. The landmark diabetes prevention program (DPP) is recognized as one of the most effective NCD prevention interventions. The DPP compared a lifestyle intervention targeting modest weight loss through dietary changes and increased PA to treatment with metformin or standard care. Target outcomes included the prevention or delay of diabetes development in pre-diabetic adults. When compared to standard care, participants in the lifestyle intervention group had a 27% reduction in the incidence of diabetes versus an 18% reduction in the metformin group, with a mean follow-up of 15 years [14]. A systematic review of DPP translational programs concluded that the DPP has been successful in its community and cultural adaptations [15]. Additionally, the DPP-like programs were found to be cost-effective among high-risk populations when delivered to groups in community or primary care settings [16].

Despite their higher risk for diabetes and cardiovascular disease, there are limited reports of lifestyle interventions in Arabic-speaking populations [17]. This study sought to evaluate a culturally adapted, community-based group lifestyle intervention based on the DPP, targeting nutrition and physical activity in Palestinian-Arab women.

## 2. Materials and Methods

The community-based lifestyle intervention (CBLI) was constructed based on publicly available material from the DPP [18]. The content was translated and modified to the specific cultural and religious sensitivities of Palestinian society. The material was piloted in a 12-session program with 38 women from EJ. This program was then further adapted into the present CBLI which consists of 20 group-based, weekly sessions of 3 h each, with 2 h of interactive lectures and one h of aerobic exercise. The final curriculum included the Mediterranean diet (MD), PA, cardiovascular health literacy as well as behavioral self-regulation techniques (goal setting, self-monitoring) and conscious eating.

This study included a 2-stage evaluation (Figure 1). Phase 1: A quasi-experimental study was conducted from June to December 2016. Two community centers in EJ (the Old City center and the Zur-Baher center) were identified as candidates for the intervention. These community centers were located in neighborhoods that were distant enough from each other to prevent treatment contamination and potential bias attributed to treatment awareness. Upon initiation of the study, baseline data were collected through self-report questionnaires and direct measurement from all participants at both sites by trained staff. The Zur-Baher community center was assigned to the intervention first due to logistic considerations and the Old City center was assigned to a delayed intervention, serving as a comparison group. Post-intervention data were collected at the final session of the 6-month intervention. For the comparison group, post-intervention data were similarly collected at the community center, after which, all participants were invited to participate in the intervention during the study’s second phase.

Phase 2: In this phase, we evaluated the dissemination of the intervention program in four community centers using a before–after program review design. Minor changes were made in the intervention as a result of feedback and process evaluation of the first phase of the study. In this phase, data were collected at baseline, after completion of the program, and at 6-month follow-up. After completion of the intervention, program managers maintained contact with the participants through group discussions using the free instant messaging mobile application “WhatsApp”, to encourage maintenance of health behaviors. Process evaluation and quality assurance in both phases were conducted through registration of attendance at each session and checklist-guided random site visits by study staff. In addition, participants provided monthly feedback on the program through a group discussion.

Recruitment consisted of inviting women who attended the community center on a regular basis to apply. Inclusion criteria included female gender, age 18 or over, and the ability to commit to the intervention timetable. Exclusion criteria included serious mental or physical illness that would hinder their ability to participate in a physical activity or nutrition program.

### 2.1. Measures

MD adherence scale items were adapted from the ATTICA study [19] including an assessment of daily average consumption of whole grains, vegetables, fruits, legumes, olive-oil and low-fat dairy products. “Western Diet” food items were assessed through a self-report questionnaire asking about consumption of sweet drinks, sweets and refined grains [20,21]. Average daily steps (ADS) were assessed in the intervention group only, using a validated pedometer (Omron Model HJ-320) [22]. Measures included ADS count over 7 days, and percentage change in ADS. Body mass index (BMI) was calculated by measuring weight (kg) on a calibrated scale (F-168, Xinfu Household Electronics, Zhongshan, Guangdong, China), and height (cm) was measured using mechanical measuring tape (ADE, MZ10017) for each participant. Blood Pressure (BP) was measured while participants were seated, using an electronic apparatus (HM-50, Visocor). BP scores were calculated by averaging two consecutive measurements, with 5 min of rest between each reading. All tests were performed on site by trained research staff including a nurse. In phase 2, PA outcomes were assessed by a questionnaire translated and culturally adapted from the Healthy Heart Score Study [23]. Questionnaires were translated by a translator whose mother tongue is Arabic and has expertise in health and medical terminology. All translations were reviewed independently by a public health professional whose mother tongue is Arabic. Outcomes included the percentage change in PA and percentage of participants who achieved 150 min of moderate PA. In the dissemination phase, blood tests for glycated hemoglobin A1C (HbA1c), LDL, triglycerides, HDL, non-HDL cholesterol were obtained by blood tests from the participants’ health plan, or using capillary whole blood obtained on finger stick through a cartridge-based latex agglutination inhibition assay (Cobas B101 point of care device, Roche Diagnostics, Basel, Switzerland).

All participants gave their informed consent for inclusion before they participated in the study. The study was conducted in accordance with the Declaration Helsinki, and the protocol was approved by the institutional review board (Helsinki Committee) of Hadassah University Medical Center (quasi-experiment: HMO-0257-13, dissemination: HMO-0585-18).

### 2.2. Statistical Analysis

Data were analyzed using the Generalized Estimating Equations (GEE) model (SPSS V.24.0, IBM Corp., 2017). The GEE model allows the utilization of a subject’s multiple responses in multiple time frames [24]. The full information of maximum likelihood approach is applied in GEE similar to its application in general linear modelling when data include missing values, thus enabling the use of all data collected, rather than losing data with missing values [25]. T-test and χ^2^ test were used to compare demographic data between groups.

## 3. Results

### 3.1. Quasi-Experiment

In total, 38 women agreed to participate in the intervention and 22 agreed to participate in the comparison group. Attendance records found that 68.5% of intervention participants attended at least 80% of the sessions. Of the participants in the experimental group, 71% completed the post-intervention data collection, and 60% of the comparison group completed the evaluation. The baseline characteristics of the participants are presented in Table 1. There was a higher percentage of married women and increased frequency of self-reported history of hypertension in the intervention group; no other statistically significant differences were seen between groups.

Table 2 presents the results of the GEE analysis. Participants in both groups increased their vegetable consumption from 1.25 servings daily at baseline to 2.01 servings post-intervention, on average. The main time effect was significant (χ^2^ = 6.11, *p* < 0.05). Increased consumption of fruit (0.73 to 1.92 average servings, *p* < 0.05), and whole grains were seen in the intervention group, as was weight reduction (−2.21 kg, *p* < 0.01). No changes were seen in the comparison group. Systolic BP was significantly lower in the intervention group at post-intervention. There was a significant increase in the ADS (from 4456 to 64,040, *p* < 0.001) as measured by pedometer in the intervention group, whereas, no data were available for the comparison group.

### 3.2. Dissemination Phase

117 women across 4 groups consented to participate in the study. Eleven participants from the comparison group in phase 1 participated in Group 1 of the dissemination phase. Baseline characteristics are presented in Table 3. Ages ranged between 19 and 71 years, and the majority of participants were 45–59 years old. Attendance of at least 80% of the sessions ranged between 60 and 83%. Overall, attendance rates observed in the dissemination phase of the intervention group ranged between 70 and 80%. Table 4 presents the GEE analysis.

## 4. Discussion

The high prevalence of obesity and diabetes among Arab women and men [26] accentuates the need for effective prevention and health promotion interventions. Yet, evidence on such interventions in this population is scarce [17]. The current study evaluated the effectiveness of a culturally adapted, group lifestyle intervention based on the DPP among Palestinian-Arab women in East Jerusalem. To our knowledge, this is the first study of a community-based lifestyle intervention outside of the health-system in this population. In the first phase, significant improvements in nutrition behaviors and weight were seen in the intervention compared to the comparison group. In both phases, there was a significant increase in the consumption of fruits and vegetables and the performance of PA as well as a reduction in weight, BP, and cholesterol. Improvements in nutrition and PA behaviors were sustained at six-month follow-up in phase 2.

The current body of literature on health behavior change in Arab women is small. In a randomized control trial of Arab women with metabolic syndrome in northern Israel, intervention participants had decreased waist circumference and weight as well as decreased prevalence of metabolic-syndrome compared to controls [27]. In other Arab countries in the Middle East, we found few publications on community-based interventions. In Saudi Arabia, a study of 250 pre-diabetic and obese women evaluated one-on-one biweekly education sessions by a nutritionist, which led to improved HbA1c and lipid parameters, but did not find a significant difference in BMI [28]. A DPP-based intensive lifestyle program in predominantly obese Saudi adults with prediabetes led to decreased weight and improved fasting glucose [29]. In a study of Emirati patients with type 2 diabetes, a lifestyle intervention improved glycemic control with gains that were maintained at one-year follow-up [30].

Our study’s findings are consistent with a previous randomized trial in Arab women in Israel [27]. Although the weight loss we found (−2.12 kg) is lower than that found in a meta-analysis of community-based DPPs [31] which specifically target a weight reduction of 5% (−2.32 kg), our intervention is notable for maintenance of weight loss and physical activity at the 6-month follow-up. Additionally, even small weight reduction is associated with health benefits. A DPP study by Hamman et al. [32] found that for every 1 kg of weight loss, there was a 16% reduction in risk for developing diabetes.

Maintenance of behavior changes were likely facilitated by multiple aspects of our intervention. Participants in both phases consistently reported that the pedometers motivated them to engage in more PA, which was apparent in participants’ increased ADS. Pedometers and similar monitoring devices have increased motivation and PA performance in people with a range of chronic conditions [33]. Participants’ PA maintenance following intervention completion (through the use of pedometers) may have contributed to weight-loss maintenance. Additionally, program managers maintained continuous contact with participants through WhatsApp following intervention completion. During this time, participants were encouraged to continue adherence to the MD and regular PA to prevent weight regain. Continued professional support is indicated as an effective method for weight-loss maintenance in several clinical trials [34]. This low-cost maintenance method required minimal time investment and likely contributed to participants’ behavior change maintenance.

Cultural adaptation was a central factor in this study, potentially improving outcomes. The process of adaptation requires far more than changing superficial aspects of an intervention such as language, food and images [35]. Given the unique culture of the targeted Palestinian community, our intervention modified multiple aspects of the DPP content based on participant feedback from the pilot intervention. For example, our nutrition module included local and inexpensive ingredients as well as substitutions and adaptations of traditional recipes. Staff and participants offered solutions for challenges to initiating nutrition changes, such as family opposition, traditional foods, and cultural norms. The culture of hospitality also presented complex challenges for this community, as it directly impacts the frequency and quantity of consumption; however, it also provided opportunities for exposing other community members to healthy nutrition options, increasing potential reach to the surrounding community. Additionally, incorporating PA into group sessions provided a convenient, safe and socially acceptable setting for PA performance that helped overcome both socio-religious and logistic barriers.

### Study Limitations and Strengths

This study used a quasi-experimental rather than randomized design due to unexpected staffing delays at the comparison center. The decision was made to proceed with the study in a quasi-experimental fashion, so as not to delay the initiation of the study. A main challenge of a quasi-experimental design is the possibility of biased outcomes due to inherent differences between groups. In this study, socio-economic parameters were similar between the groups, suggesting that differences in social determinants of health did not influence outcome differences.

Attrition rates were very high in our comparison group follow-up in phase 1. This is a quite common phenomenon in naturalistic studies, particularly in groups that did not receive an intervention [36]. As a naturalistic study, our data collection was also somewhat challenged. The phase 1 comparison group’s follow-up data were collected at the beginning of the phase 2 intervention rather than at the end of phase 1’s intervention, creating a discrepancy in the collection time of post-data between the intervention and comparison groups. Given the logistic and staffing challenges, this timing was necessary in order to allow for complete data collection. The study was also limited by self-report measures of nutrition and PA; however, this type of data collection is often the only feasible method in community-based studies.

This study provides evidence for the feasibility of applying a community-based, culturally adapted DPP with Palestinian-Arab women, with improved outcomes in nutrition and PA. The long and careful process of program adaptation to the community’s familial and socio-cultural needs likely contributed to the program’s success. Additionally, the group setting contributed to a warm and supportive atmosphere, likely increasing participants’ engagement and motivation. This was reflected in the high weekly attendance rate and modest drop-out rates in both phases. In addition, collaborative efforts with the local community centers and consistent support from the study staff likely facilitated behavior change maintenance at the 6-month follow-up assessment.

## 5. Conclusions

This study describes a culturally adapted, community-based lifestyle intervention targeting high risk women that led to participants’ improved nutrition and PA. Collaboration with community centers created an effective channel for intervention dissemination, increasing community member access. Future research is warranted to better understand the challenges and opportunities for changing and maintaining positive health behaviors in this high-risk population as well as to assess long-term effectiveness. Cost-effectiveness analyses would also inform the scalability and feasibility of these interventions in similar low-resourced communities.

## Figures and Tables

**Figure 1 ijerph-17-09184-f001:**
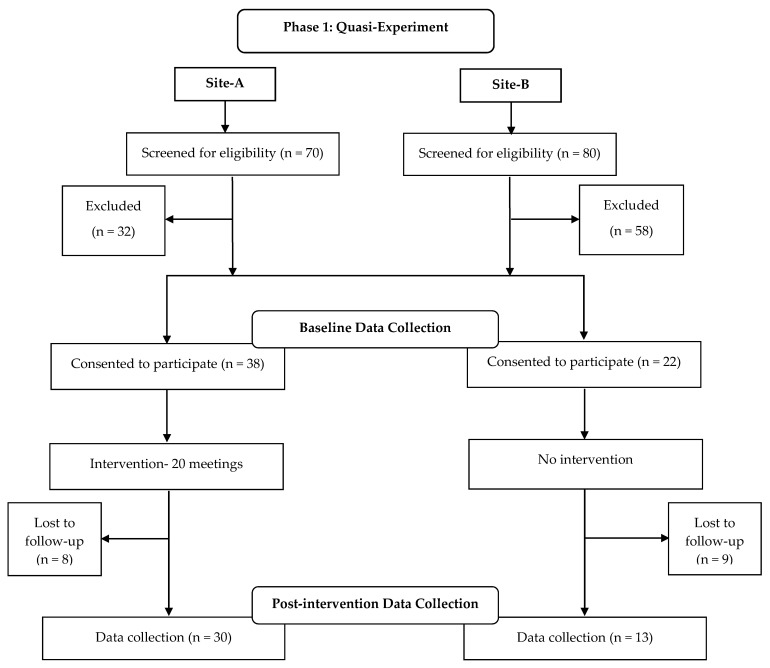
Flow chart of the study process.

**Table 1 ijerph-17-09184-t001:** Baseline sociodemographic and health characteristics of participants in the quasi-experimental study.

Characteristics	Intervention (*n* = 38)	Comparison (*n* = 22)	^1^*p*-Value
Age, Mean ± SD	48.4 ± 12.7	57 ± 11.3	0.12
Education level; *n* (%)			0.08
Elementary school	9 (23.7%)	10 (45.5%)	
High school	22 (57.9%)	8 (36.4%)	
Matriculation exam	7 (18.5%)	4 (18.6%)	
Marital status; *n* (%)			0.006
Married	34 (89.5%)	13 (59.1%)	
Other	4 (10.5%)	9 (40.9%)	
Parity, Mean (SD)	5.3 (2.3)	5.4 (2.1)	0.923
Employed *n* (%)	6 (16.2%)	1 (4.5%)	0.24
Self-reported HOD, *n* (%)			
Hypertension	4 (11.4%)	13 (59.1%)	<0.001
Hypercholesterolemia	14 (40.0%)	13 (59.1%)	0.16
Diabetes	6 (16.2%)	1 (4.5%)	0.18
Health behaviors: smoker, *n* (%)	7 (18.9%)	5 (22.7%)	0.73
BMI categories, *n* (%)			0.13
Normal	6 (15.8%)	0	
Overweight	14 (36.8%)	11 (52.4%)	
Obese	18 (47.4%)	10 (47.6%)	
SBP, Mean (SD), mmHg	129.6 (17.2)	126.5 (13.7)	0.466
DBP, Mean (SD), mmHg	77.2 (9.7)	79.9 (11.1)	0.333

^1^*p*-value for two independent sample *t*-test if continuous, and χ^2^ test if categorical. Abbreviations, HOD: history of disease, BMI: body mass index, SBP: systolic blood pressure, DBP: diastolic blood pressure.

**Table 2 ijerph-17-09184-t002:** Generalized Estimating Equations (GEE) regression analysis for differences in outcomes between baseline and six months for the quasi-experiment.

	Time			Group			Time × Group				
Dependent Variable	Wald ^c^	Before	After	Wald	Control	Intervention	Wald	Control Before	Control After	Intervention Before	Intervention After
Average daily portions of:											
Vegetables	6.11 *	1.25 ^a^ (0.16)	2.01 ^b^ (0.26)	0.05	1.60 ^a^ (0.22)	1.66 ^a^ (0.19)	2.64 ~	1.47 ^a^ (0.23)	1.58 ^a^ (0.41)	1.15 ^a^ (0.21)	2.23 ^b^ (0.32)
Fruits	6.02 *	1.33 ^a^ (0.14)	2.00 ^b^ (0.25)	7.73 **	2.05 ^b^ (0.25)	1.28 ^a^ (0.15)	7.99 **	2.09 ^a^ (0.28)	1.68 ^a^ (0.43)	0.73 ^a^ (0.10)	1.92 ^b^ (0.27)
Vegetables and fruits	6.77 **	2.57 ^a^ (0.25)	3.92 ^b^ (0.47)	1.84	3.58 ^a^ (0.43)	2.91 ^a^ (0.30)	7.51 **	3.56 (0.42)	3.14 (0.70)	1.86 ^a^ (0.26)	4.10 ^b^ (0.57)
Whole grains	0.07	0.8 (0.13)	0.85 (0.14)	1.2	0.7 (0.11)	0.93 (0.16)	6.2 *	0.9 ^a^ (0.15)	0.4 ^b^ (0.08)	0.8 ^a^ (0.2)	1.1 ^a^ (0.22)
Average daily Steps	21.34 ***	4456 ^a^ (388)	6404 ^b^ (557.3)	-	-	-	-	-	-	-	-
SBP (mmHG)	8.80 **	128.82 ^b^ (2.07)	121.71 ^a^ (2.23)	0.45	126.48 ^a^ (2.89)	124.05 ^a^ (2.15)	21.17 ***	126.5 ^a^ (2.86)	130.05 ^a^ (3.22)	129.66 ^b^ (2.76)	117.06 ^a^ (2.48)
DBP (mmHG)	4.33 *	78.54 ^a^ (1.39)	73.86 ^b^ (2.38)	0.52	77.27 ^a^ (2.81)	75.12 ^a^ (1.26)	0.06	79.95 ^a^ (2.32)	74.28 ^a^ (5.71)	77.26 ^a^ (1.57)	73.09 ^a^ (1.37)
SBP ≥ 140 (mmHG) ^c^	4.39 *	0.22 ^b^ (0.06)	0.09 ^a^ (0.06)	0.29	0.16 ^a^ (0.08)	0.12 ^a^ (0.04)	1.68	0.23 ^a^ (0.09)	0.17 ^a^ (0.09)	0.21 ^a^ (0.07)	0.04 ^a^ (0.04)
Weight (kg)	9.47 **	79.66 ^b^ (1.96)	78.64 ^a^ (1.95)	0.30	80.21 ^a^ (2.99)	78.08 ^a^ (2.50)	2.26	80.42 ^a^ (3.01)	80.44 ^a^ (2.98)	78.74 ^a^ (2.52)	77.34 ^a^ (2.50)
BMI	10.18 **	31.19 ^b^ (0.69)	30.78 ^a^ (0.68)	0.13	31.23 ^a^ (0.98)	30.73 ^a^ (0.95)	2.11	31.32 ^a^ (1.00)	31.30 ^a^ (0.96)	30.99 ^a^ (0.96)	30.44 ^a^ (0.95)

Abbreviations, SBP: systolic blood pressure, DBP: diastolic blood pressure, BMI: body mass index. For significant effects, Latin letters show upward ranking, while “^a^” represents the lowest estimate and “^b^” represents the highest estimate. These estimations were based on a post-hoc pairwise comparison with Bonferroni correction. “~” stands for rejection criterion at *p* < 0.10. ^c^ For dichotomous dependent variables, the marginal means are estimated probabilities. *p*-value: *** *p* < 0.001, ** *p* < 0.01, * *p* < 0.05. Cronbach’s Alpha: 0.95.

**Table 3 ijerph-17-09184-t003:** Baseline sociodemographic and health characteristics of participants in the dissemination phase.

Variable	*n* = 117
Age group *n* (%)	
<45	44 (37.9%)
45–59	54 (46.6%)
60+	18 (15.5%)
Education level: *n* (%)	
Elementary school	14 (12.1%)
High school	53 (45.7%)
Vocational training	27 (23.3%)
Academic degree	22 (19%)
Marital status; *n* (%)	
Married	92 (79.3%)
Other	24 (20.7%)
Parity, Mean (SD)	4.18 (2.3%)
Participant’s employment	
Yes	8 (7%)
No	107 (93%)
Spouse employed	77 (81.1%)
Self-reported HOD, *n* (%)	
Hypertension	20 (17.1%)
Heart attack	3 (2.6%)
Hypercholesterolemia	39 (33.3%)
Diabetes	22 (18.8%)
Weight, Mean (SD), kg	77.9 (16.0%)
BMI categories: *n* (%)	
Normal	18 (5.5%)
Overweight	39 (33.6%)
Obese	58 (50%)
SBP, Mean (SD), mmHg	128.6 (20%)
DBP, Mean (SD), mmHg	81 (14.8%)

Abbreviations, HOD: history of disease, BMI: body mass index. SBP: systolic blood pressure, DBP: diastolic blood pressure.

**Table 4 ijerph-17-09184-t004:** GEE regression analysis for differences in outcomes between baselines, six months, and 6 months following the program completion for the dissemination phase.

Dependent Variable	Wald Chi-Square			Marginal Means			
	Time	Group	Time × Group	Time 1	Time 2	Time 3	Total
Average daily portions of:							
Vegetables	12.01 **	11.06 **	11.01 *	1.76 ^a^ (0.11)	2.32 ^b^ (0.13)	2.09 ^b^ (0.14)	2.06 (0.08)
Fruit	0.62	4.57	5.02	1.81 (0.12)	1.91 (0.09)	1.89 (0.12)	1.87 (0.08)
Vegetables & Fruit	7.30 *	9.63 **	6.20	3.60 ^a^ (0.20)	4.25 ^b^ (0.18)	3.99 ^a b^ (0.22)	3.95 (0.14)
Achieving the recommended ≥ 5 per day	10.88 **	1.81	2.58	0.25 ^a^ (0.04)	0.43 ^b^ (0.05)	0.44 ^b^ (0.06)	0.37 (0.03)
Whole grains	1.33	31.4 ***	11.97 *	1.23 (0.12)	1.38(0.10)	1.32 (0.10)	1.31 (0.07)
Sweetened drinks	2.66	6.41 *	22.46 ***	2.66	0.65 (0.12)	0.85 (0.17)	0.54 (0.10)
Sweets	9.48 **	1.20	17.82 **	9.48 **	1.48 ^a^ (0.13)	1.03 ^b^ (0.11)	1.20 (0.14)
Olive oil (tbl per week)	0.41	5.59 *	7.71 *	4.68 (0.33)	4.77 (0.43)	4.98 (0.47)	4.81 (0.29)
Average daily steps (ADS)	63.69 ***	14.99 **	1.36	4087.4 ^a^ (188.6)	5284.2 ^b^ (234.4)	-	4647.4 (196.6)
Participants increased their ADS > 4500	26.70 ***	8.70 *	0.50	0.36 ^a^ (0.05)	0.64 ^b^ (0.05)	-	0.50 (0.05)
PA moderate intensity	22.78 ***	1.75	4.92	97.82 ^a^ (13.21)	191.32 ^b^ (19.97)	183.95 ^b^ (28.40)	151.00 (13.68)
PA vigorous intensity	12.10 **	4.02	15.36 **	99.12 ^a^ (16.16)	155.72 ^b^ (24.50)	99.92 ^a^ (22.28)	104.59 (15.27)
Achieving the recommended PA > 150 min/week	24.17 ***	0.04	8.07	0.33 ^a^ (0.05)	0.64 ^b^ (0.05)	0.53 ^b^ (0.6)	0.50 (0.04)
SBP	22.54 ***	3.40	10.84 *	128.15 ^b^ (1.88)	121.34 ^a^ (1.66)	119.01 ^a^ (2.12)	122.77 (1.44)
DBS	11.52 **	3.56	2.24	80.80 ^b^ (1.40)	77.03 ^a^ (0.95)	75.46 ^a^ (1.69)	77.73 (1.00)
SBP ≥ 140	7.66 *	0.85	6.49	0.27 ^b^ (0.04)	0.19 ^b^ (0.04)	0.08 ^a^ (0.05)	0.16 (0.04)
Weight	1.10	1.05	10.20 *	77.52 (1.45)	76.88 (1.41)	76.81 (1.42)	77.07 (1.38)
BMI	1.64	0.18	11.60 *	30.41 (0.52)	30.11 (0.50)	30.18 (0.52)	30.24 (0.50)
HbA1C	3.25	3.03	2.29	5.71 (0.10)	5.67 (0.08)	5.34 (0.22)	5.57 (0.09)
Total cholesterol mg/dL	7.91 **	0.51	0.80	187.07 ^b^ (3.05)	176.33 ^a^ (3.81)	-	181.62 (2.91)
HDL-mg/dL	2.58	5.40	1.36	51.30 (1.49)	47.98 (1.66)	-	49.61 (1.20)
LDL-mg/dL	7.49 **	2.47	1.36	111.55 ^b^ (2.84)	97.58 ^a^ (4.22)	-	104.33 (2.68)
TG-mg/dL	0.45	0.54	9.09 *	123.20 (5.97)	118.36 (7.76)	-	12.75 (5.94)

Abbreviations, tbl: table spoon, PA: physical activity, SBP: systolic blood pressure, DBP: diastolic blood pressure, BMI: body mass index, glycated hemoglobin A1C. For significant effects, Latin letters show upward ranking, while “^a^” represents the lowest estimate and “^b^” represents the highest estimate. “^a b^” indicates no change in the estimate at Time 3. These estimations were based on a post-hoc pairwise comparison with Bonferroni correction. For dichotomous dependent variables, the marginal means are estimated probabilities. *p*–value: *** *p* < 0.001, ** *p* < 0.01, * *p* < 0.05. Cronbach’s Alpha: 0.95.

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
