# Peer review of "Two-Phase Evaluation of a Community-Based Lifestyle Intervention for Palestinian Women in East Jerusalem: A Quasi-Experimental Study Followed by Dissemination"

_ijerph, 2020, doi:10.3390/ijerph17249184_

Round 1

Reviewer 1 Report

In this article, there are some shortcomings and questions.

  • In my opinion, this paper is primarily relevance descriptions of data, and lack of novelty.
  • It is well-known that fruit, vegetables and whole grains, weight reduction can decrease blood pressure and total cholesterol and LDL.
  • The samples are too small. Please increase the number of samples.
  • The experiments do not have right control group. Season, climate, and economic condition also be potential factors in the 6 months.
  • English language is required to be improved.

Author Response

In my opinion, this paper is primarily relevance descriptions of data, and lack of novelty.

Authors’ response:We are pleased that reviewer recognized the extensive data presented in this manuscript.  We respectfully disagree that the study lacks novelty.  The originality in this study arises from the unique population, the lack of similar interventions despite the high morbidity rates of diabetes and CVD’s, the extensive cultural adaptation that was required, the inclusion of a broad representation of community members in the intervention rather than targeting a pre-diabetic population, and the challenges of implementation in a non-clinical community setting.

It is well-known that fruit, vegetables and whole grains, weight reduction can decrease blood pressure and total cholesterol and LDL.

Authors’ response:  We agree with the reviewer that changes in nutrition behaviors and weight reduction are associated with reduction in blood pressure and LDL. We believe that it is a strength of this study that we were able to demonstrate statistically and clinically significant changes in these anthropometric parameters that suggest that the changes enabled by this intervention will translate into clinically meaningful improvements in health and prognosis.

The samples are too small. Please increase the number of samples.

Authors’ response: We agree that this is a potential limitation of the study. This study describes 6 separate community-based interventions involving 60 women in 2 groups in the quasi-experimental phase, and 117 women in four groups in the dissemination phase.  Using this structure, we were able to demonstrate statistically significant and meaningful changes in nutrition behaviors, steps as measured by pedometer, and BMI. Given the major efforts required to perform community-based interventions, we believe that at this stage, this study makes a valuable contribution to the field.

The experiments do not have right control group. Season, climate, and economic condition also be potential factors in the 6 months.

Authors’ response: The reviewer’s concern about season and climate is well-taken.  It is possible that differences in fruit and vegetable consumption were related to seasonal change. We do believe, however, that it is unlikely that differences in whole grain consumption, BMI and blood pressure were related to seasonal differences over a 6-month time period. As such, we are confident that our results are valid and that the differences between the intervention and the comparison group are related to the intervention.

English language is required to be improved.

Authors’ response: The manuscript has been edited by a native speaker of English with extensive experience in editing.

Reviewer 2 Report

Please go thoroughly throughout the guide for authors of International Journal of Environmental Research and Public Health and make necessary technical corrections to your manuscript, e.g. font face, font size, spacing of paragraphs, gaps, dots, commas, etc.):

Line 15: Abstract – correct Abstract:

Line 36: with – correct With

Line 77, 98, 222 - two gaps ??

Line 92-93: The intervention underwent minor revisions that were informed by phase one outcomes. - I don't understand this sentence.

Line 114: measuring of weight (kilograms) on a calibrated scale and height (cm) - please complete with the type and the manufacturer

Line 115: electronic apparatus - please complete with the type and the manufacturer

Line 126: the protocol was approved by the institutional review board (Helsinki Committee) of Hadassah University Medical Center - please add the protocol identification number

Line 145: the values indicated in Table 1 –  are missing % ???

Line 153: -2.21  - unit is missing?

Line 158: Table 2 – the indicated values - different font size

Line 209: up[30] – correct up [30] – gap between words

Line 212: -2.32 kg – correct -2.32kg - without gap

Line 215: of weight which is associated – correct of weight is associated ???

Please provide a certificate with the date that grammar was improved in the revised manuscript.

This study addresses an interesting research question. However, I think there is much that could be improved on in terms of design, reporting and interpretation.

Author Response

Please go thoroughly throughout the guide for authors of International Journal of Environmental Research and Public Health and make necessary technical corrections to your manuscript, e.g. font face, font size, spacing of paragraphs, gaps, dots, commas, etc.):

Line 15: Abstract – correct Abstract:

Line 36: with – correct With

Line 77, 98, 222 - two gaps??

Line 114: measuring of weight (kilograms) on a calibrated scale and height (cm) - please complete with the type and the manufacturer

Line 115: electronic apparatus - please complete with the type and the manufacturer

Line 126: the protocol was approved by the institutional review board (Helsinki Committee) of Hadassah University Medical Center - please add the protocol identification number

Line 145: the values indicated in Table 1 –  are missing % ???

Line 153: -2.21  - unit is missing?

Line 158: Table 2 – the indicated values - different font size

Line 209: up[30] – correct up [30] – gap between words

Line 212: -2.32 kg – correct -2.32kg - without gap

Line 215: of weight which is associated – correct of weight is associated ???

Please provide a certificate with the date that grammar was improved in the revised manuscript.

Authors’ response: We have made the changes detailed above. The manuscript was reviewed on November 25th by a native speaker of English with extensive experience in editing.

Line 92-93: The intervention underwent minor revisions that were informed by phase one outcomes. - I don't understand this sentence.

Authors’ response:  This sentence has been changed to: “Minor changes were made in the intervention as a result of feedback and process evaluation of first phase of the study.” 

This study addresses an interesting research question. However, I think there is much that could be improved on in terms of design, reporting and interpretation.

Authors’ response:  We appreciate the interest of the reviewer in our study, and we agree with the reviewer that a randomized controlled study would have been preferable to the quasi-experimental design of this study. Given the exigencies of interventions in the community setting, we opted to begin the implementation of the study when circumstances permitted rather than waiting for ideal conditions for randomization.  We would be interested in all further specific suggestions of the reviewer as to how to improve the reporting and interpretation of our data.

Reviewer 3 Report

The paper reports a study that addresses one the growing public health concern globally. I think the study is relevant and unique given the study population, and the fact such studies are not quite common. The paper introduction is adequate, clear, and straightforward. The discussion is adequate.

The following are some observations about the study design/methodology that authors might consider addressing:

Authors should explain why they invited the comparison group in the pilot quasi-experimental phase to join the second phase (dissemination stage).  What are the potential impacts of the comparison group who joined the phase two on the outcomes? Is it not a limitation?  

Another question is why comparison group only for the pilot study (phase 1), but not for dissemination phase as well?

On line 210, authors indicated “we observed a modest reduction in mean weight at 6-months follow-up” Was this reduction observe in both phases or only in phase 1?

Author Response

The paper reports a study that addresses one the growing public health concern globally. I think the study is relevant and unique given the study population, and the fact such studies are not quite common. The paper introduction is adequate, clear, and straightforward. The discussion is adequate.

The following are some observations about the study design/methodology that authors might consider addressing:

Authors should explain why they invited the comparison group in the pilot quasi-experimental phase to join the second phase (dissemination stage).  What are the potential impacts of the comparison group who joined the phase two on the outcomes? Is it not a limitation?  

Authors’ response: We appreciate this thoughtful comment.  We opted for a “wait-list” comparison, due to the ethical advantage that it allows for provision of the intervention to research participants who are seeking help, whilst permitting a non-intervention evaluation. The option to participate in the intervention considerably improved our ability to collect data from the comparison group.  As this study was performed in the community rather than in a medical or structured setting, this advantage is significant. Given the overall stability of findings within the wait list control, we do not believe that the initial data collection itself was an intervention that would lead to lack of response to the intervention itself.  It is possible that these women were particularly interested and committed to completing the intervention, which may have improved the findings of the group overall.  As these women participated in the dissemination phase rather than a second comparison study, and are such a small component of the dissemination study, we do not believe that this is a significant limitation to the study.

Another question is why comparison group only for the pilot study (phase 1), but not for dissemination phase as well?

Authors’ response: This point is well taken. The addition of additional comparison groups would have strengthened the study. We found that the wait-list control method over the 6 months of the study was extremely challenging, and we were convinced that there was a substantial risk that attrition rates would jeopardize the validity of the study.

On line 210, authors indicated “we observed a modest reduction in mean weight at 6-months follow-up” Was this reduction observe in both phases or only in phase 1?

Authors’ response: This reduction was observed in phase 2. We did not conduct a 6 month follow up for the quasi-experimental phase, and thus these data are not available.

Round 2

Reviewer 1 Report

In this study, the author investigates that the intervention on lifestyle habits can improved health parameters which is related to diabetes and cardiovascular.
It has certain practical importance.